# Neuron-specific *Agrin* splicing by Nova RNA-binding proteins regulates conserved neuromuscular junction development in chordates

Md. Faruk Hossain[1], Sydney Popsuj[2], Burcu Vitrinel[3], Nicole A. Kaplan[3], Alberto Stolfi [2]*, Lionel Christiaen[3,4]*, Matteo Ruggiu[1]*

1 Laboratory of RNA Biology & Molecular Neuroscience, Department of Biological Sciences, St. John's University, Queens, New York, United States of America, 2 School of Biological Sciences, Georgia Institute of Technology, Atlanta, Georgia, United States of America, 3 Department of Biology, New York University, New York, New York, United States of America, 4 Michael Sars Centre, University of Bergen, Bergen, Norway

* alberto.stolfi@biosci.gatech.edu (AS); Lionel.Christiaen@uib.no (LC); ruggium@stjohns.edu (MR)

## Abstract

In mammals, neuromuscular synapses rely on clustering of acetylcholine receptors (AChRs) in the muscle plasma membrane, ensuring optimal stimulation by motor neuron-released acetylcholine neurotransmitter. This clustering depends on a complex pathway based on alternative splicing of *Agrin* pre-mRNAs by the RNA-binding proteins Nova1/2. Neuron-specific expression of Nova1/2 ensures the inclusion of small "Z" exons in *Agrin*, resulting in a neural-specific form of this extracellular proteoglycan carrying a short peptide motif that is required for binding to Lrp4 receptors on the muscle side, which in turn stimulate AChR clustering. Here we show that this intricate pathway is remarkably conserved in *Ciona robusta,* a non-vertebrate chordate in the tunicate subphylum. We use *in vivo* tissue-specific CRISPR/Cas9-mediated mutagenesis and heterologous "minigene" alternative splicing assays in cultured mammalian cells to show that *Ciona* Nova is also necessary and sufficient for *Agrin* Z exon inclusion and downstream Lrp4-mediated AChR clustering. We present evidence that, although the overall pathway is well conserved, there are unexpected differences in Nova structure-function between *Ciona* and mammals. We further show that, in *Ciona* motor neurons, the transcription factor Ebf is a key activator of *Nova* expression, thus ultimately linking this RNA switch to a conserved, motor neuron-specific transcriptional regulatory network.

## Introduction

The human brain contains about 86 billion neurons [1]. For the brain to function, neurons have to communicate with each other via connections called synapses,

**Data availability statement:** All relevant data are within the paper and its Supporting information files.

**Funding:** This work was supported by grants R35GM158421 from NIH/NIGMS, R01HD104825 from NIH/NICHD, and 1940743 from NSF/IOS to AS, grants R01GM96032, R01HL108643, and R01HD096770 from NIH to LC, and grant R15GM119099-01 from NIH/NIGMS to MR. The content is solely the responsibility of the authors and does not necessarily represent the official views of the National Institutes of Health. The funders had no role in study design, data collection and analysis, decision to publish, or preparation of the manuscript.

**Competing interests:** The authors have declared that no competing interests exist.

**Abbreviations:** AChR, acetylcholine receptor; CMS, congenital myasthenic syndrome; hpf, hours post-fertilization; ISH, in situ hybridization; MG, Motor Ganglion; MN2, Motor Neuron 2; NISE, Nova-binding intronic splicing enhancer element.

and a single neuron can have tens of thousands of synapses [2]. This staggering architectural and circuitry complexity is essential to process sensory information and control the body's response to external stimuli and, ultimately, cognition and behavior [3,4]. At the same time, such complexity makes it difficult to study individual synapses, particularly at the molecular level. Due to its large size and experimental accessibility, the neuromuscular junction (NMJ), a peripheral cholinergic synapse between a motor neuron and a muscle cell [3,5–7], is arguably the best-understood mammalian synapse, and NMJ deterioration is at the center of the neuromuscular disorder amyotrophic lateral sclerosis [8–12]. Motor neurons secrete a large extracellular proteoglycan named Agrin for its ability to promote aggregation of acetylcholine receptor (AChR) clusters on the muscle surface [13–15]. Agrin is synthesized by most cells of the body, but only neurons produce an alternatively spliced isoform of *Agrin* termed Z+ (or neural) *Agrin*. The two Z microexons encode a short domain of 8–19 amino acids that confers up to a 1,000-fold increase in AChR clustering activity compared to Z-negative Agrin, the isoform that does not include the Z exons [15–21]. In fact, *Agrin* KO mice and mice in which the Z exons have been deleted both die at birth from diaphragmatic paralysis [15,20], suggesting that the Z exons are essential for Agrin function. Interaction of Z+ Agrin with its postsynaptic receptor Low-density lipoprotein receptor-related protein 4 (Lrp4) [22–24] leads to the phosphorylation of the muscle-specific receptor tyrosine kinase MuSK, and, through a cascade of events, it induces AChR clustering on the muscle [15,25–28]. Defects in this pathway are responsible for many cases of the congenital neuromuscular disorder Congenital Myasthenic Syndrome, or CMS [29–39].

Despite the central role of *Agrin* Z exons in synapse biology, only recently have we started to understand how Z exon splicing is regulated. The neuron-enriched splicing factors NOVA1 and NOVA2 underlie an autoimmune neuromuscular disorder [40,41], and *NOVA2* mutations cause a severe form of neurodevelopmental disorder [42]. Double knockout mice for both *Nova1* and *Nova2* fail to include the Z exons of *Agrin* [43]; however, as Nova proteins regulate about 700 alternative splicing events in the brain [44], whether this is a direct effect remains unclear. How Nova proteins bind to and promote *Agrin* Z exon inclusion is still largely unknown.

To elucidate the role of Nova in regulation of *Agrin* Z exon splicing, we focused on the tunicate *Ciona robusta*. Tunicates, or sea squirts, are the closest living relatives to vertebrates within the chordate phylum [45,46]. The central nervous system (CNS) of the *Ciona* larva contains only 177 neurons [47], and its connectome has been completed [47–50]. Yet this minimal nervous system is formed and compartmentalized in a manner similar to that of the larger nervous systems of vertebrates [51]. This relative simplicity, alongside rapid development and a compact genome that has not undergone duplications seen in vertebrates, makes *Ciona* uniquely suited to dissect the evolutionary biology of protein-RNA regulatory switches that are important for synapse biology and neurologic disorders. In this work we show that the motor neuron terminal selector Ebf [52] activates the transcription of *Nova*, which is present as a single-copy gene in *Ciona*. Nova protein in the larval motor neurons directly promotes the inclusion of *Agrin* Z exons, which in turn stimulates acetylcholine clustering

at the NMJ through Lrp4 receptors, just as in vertebrates. By elucidating this splicing event at the molecular level, we uncover unexpected features of Nova that contribute to its function as a splicing factor. We also provide evidence of coevolution of Nova and the regulatory sequences embedded in the *Agrin* pre-mRNA that mediate Nova-dependent splicing, revealing "developmental system drift" of an otherwise highly conserved RNA splicing-dependent molecular switch.

## Results

### Identification of divergent Z exons in *Ciona robusta Agrin*

Previous bioinformatic analysis of potential Nova splicing targets in *Ciona* and other invertebrates did not indicate *Agrin* as a potential target, suggesting that *Agrin* Z exon splicing regulation by Nova was a vertebrate-specific innovation [53,54]. However, we identified two cryptic exons in between annotated exons 40 and 41 (Fig 1A and 1B), which were confirmed by Sanger sequencing of clones obtained from mixed larval and adult cDNA libraries (see S1 File). We named these microexons "Z6" and "Z5" as they are predicted to encode 6 and 5 amino acid-long polypeptide sequences, respectively (Fig 1C). Exon Z6 in particular encodes an N-X-F motif that might be functionally equivalent to the N-X-I/V motif that is encoded by the Z8 exon (exon 32) of mammals and mediates the interaction between neural Agrin and Lrp4 [55,56]. Through predicted protein sequence alignments, we found that the corresponding motif in the related species *C. savignyi* is N-X-V, supporting the idea that these sequences are likely to be conserved, functional motifs for Lrp4 binding, encoded by homologous Z exons. Using AlphaFold Server [57], we predicted the N-X-F motif of *C. robusta* Agrin (Z6 isoform) to interact with conserved residues in the same binding site of Lrp4 as previously determined by structural analysis of mammalian Agrin (Z8) and Lrp4 [55,58](S1 Fig). A similar microexon encoding an N-X-I motif in the same position was identified in a cephalochordate (amphioxus) *Agrin* gene model from *Branchiostoma lanceolatum* (S2 Fig). In contrast, we did not find evidence for any N-X-I/V/F motifs encoded by potential microexons in nematode or sea urchin *Agrin* genes (S3 Fig), suggesting that the Z exons in *Agrin* are a chordate-specific innovation.

To determine when the Z exons are included in the *Ciona Agrin* mRNAs during development, we performed a time-series of RT-PCR using primers specifically designed to amplify the region encoded by the Z6 or Z5 exons (Fig 1D). Although *Agrin* transcripts were detected in unfertilized eggs and early embryonic stages, Z exon-specific amplicons were only detected starting around 10 h post-fertilization (hpf) at 20 °C (~stage 22, or mid-tailbud II), continuing through larval stages. "Z11" *Agrin* transcripts (containing both Z6 and Z5 exons) were detected from 10 hpf (stage 22) onwards, including in the adult brain but not in the heart (Fig 1D). These data suggest that *Agrin* Z exon inclusion is occurring primarily in neural tissue, and during neuronal differentiation in embryogenesis.

### Developmental regulation of *Nova* and *Agrin* expression in the *Ciona* embryo

To determine whether *Nova* is expressed at the same time when we observe Z exon inclusion in *Agrin* transcripts, we performed a similar RT-PCR time-series for the single ortholog of mammalian *Nova1*/*Nova2* in *C. robusta*. This gene, which we call simply "*Nova*," appears to encode two major isoforms that differ in their first exon (Fig 1E). Transcripts including the more 5′ first exon (exon "1a") encode an isoform of the Nova protein that includes a predicted N-terminal nuclear localization signal (NLS). In contrast, those including the more 3′ first exon (exon "1b") do not appear to encode an NLS. We termed these two isoforms "MMM" and "MLN" (Fig 1F), respectively, based on the first three amino acid residues of their protein sequences. By RT-PCR we detected both isoforms as early as the unfertilized eggs, though the "MLN" isoform is the most abundant one at this stage (Fig 1G). Both transcript variants were expressed throughout embryogenesis and in the adult stage, though expression is higher in the brain (cerebral ganglion) than in the heart. In larvae and adult brains, both isoforms seemed to be equally abundant (S4 Fig).

*Nova* expression during *Ciona* development was previously investigated using whole-mount mRNA in situ hybridization (ISH) and reported as specific to the CNS starting at the neurula stage onwards [53]. However, the exact identities of Nova-expressing cells were not reported. Therefore, we decided to characterize *Nova* expression in greater detail by ISH

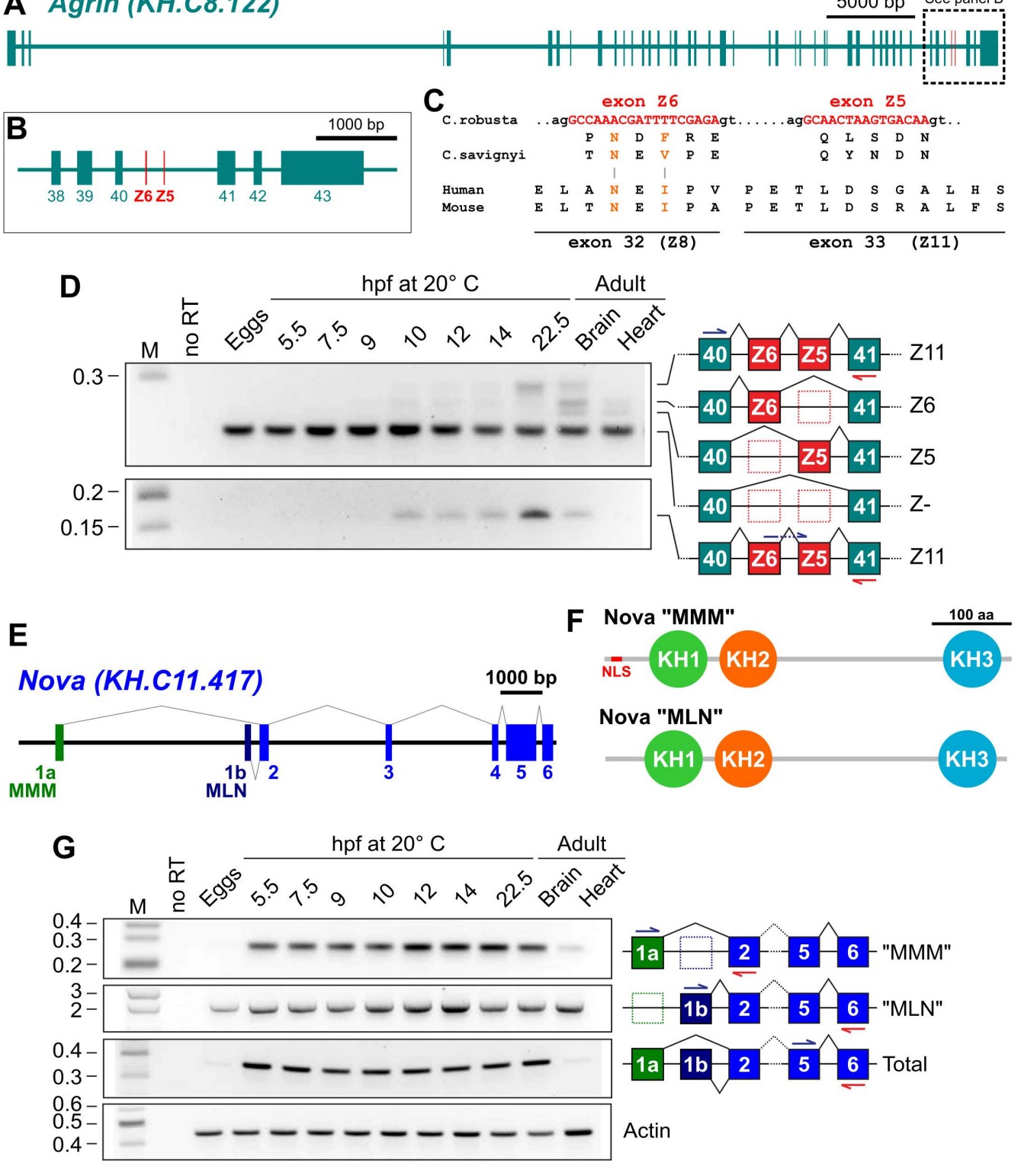

**Fig 1. Agrin and Nova expression and splicing in Ciona robusta development. (A)** Diagram of *Agrin* gene (KyotoHoya gene model ID: KH.C8.122) in *C. robusta* showing exons as thicker rectangles. **(B)** Zoomed in view of dashed region spanning constitutive exons 38–43 of *Agrin*, indicating the position of the Z exons (Z6 and Z5). **(C)** Predicted DNA coding sequences (top) of *C. robusta* Z6 and Z5 exons of *Agrin*, showing predicted protein

sequences underneath, aligned to corresponding *Ciona savignyi* protein sequences. At bottom, protein sequences encoded by the Z8 and Z11 exons of mouse and human *Agrin*, for comparison. Proposed conserved NXI/V/F peptide motif highlighted in orange font. **(D)** RT-PCR gel profiling Z exon inclusion in *C. robusta Agrin* mRNAs extracted from different developmental stages and two different adult tissues. Top gel performed with primers specific to flanking constitutive exons 40 and 41, which can simultaneously amplify Z-negative and different $Z^+$ isoforms (Z5, Z6, Z11). Bottom gel performed using a forward primer spanning the Z5/Z6 exon-exon junction, which amplifies only the Z11 isoform. **(E)** Diagram of *Nova* gene (gene ID KH.C11.417) in *C. robusta,* indicating the two alternative 1st exons (1a and 1b) encoding the Nova proteins starting with the peptides MMM and MLN, respectively. **(F)** Diagrams of the predicted protein domain organization "MMM" and "MLN" isoforms of Nova, showing the N-terminal NLS present only in the MMM isoform. **(G)** RT-PCR profiling of *Nova* expression and alternative splicing, using primers amplifying either MMM, MLN, or all Nova isoforms. Cellular actin transcripts used as a positive control for RT-PCR. M: DNA molecular weight marker in kilobase pairs. no RT: no reverse transcriptase added. Embryonic developmental stages given in hours post-fertilization (hpf) at 20 °C.

and reanalysis of recent single-cell RNA sequencing (scRNAseq) data [59]. By ISH, we first detected *Nova* transcription in neural progenitors at the early gastrula stage (Fig 2A). *Nova* transcription continued in neural progenitors in the neural plate at late gastrula (Fig 2B), and subsequently throughout the neural tube in early mid-tailbud stage embryos (Fig 2C). At this stage we also noticed expression in non-neural tissues: a subset of posterior mesenchyme cells (Figs 2C–2E and S5A), the cardiopharyngeal mesoderm (i.e., trunk ventral cells, or TVCs, Fig 2C), and very weakly in oral siphon muscle precursors (Fig 2E). The cardiopharyngeal mesoderm staining confirms earlier reports of *Nova* expression in this lineage by microarray and scRNAseq profiling [59–63].

In later tailbud embryos (~stage 24), we detected *de novo* upregulation of *Nova* transcripts in specific left/right pairs of cells in the motor ganglion (MG), which are differentiating motor neurons (Fig 2D and 2E). First, at 15 hpf at 18 °C, *Nova* was upregulated in a pair of posterior cells (Fig 2D). Slightly later (16 hpf, 18 °C), *Nova* transcripts were observed in at least two pairs of MG cells (Fig 2E). In these cells, we observed upregulation of *Nova* as a strong pulse of stained transcripts localized primarily to the cell nucleus. *Nova* expression continued in the MG, brain, and siphon muscle precursors during the larval stage (Fig 2F).

To precisely identify the *Nova*-expressing cells in the MG, we performed double ISH for *Nova* and *Islet* [64,65], a known marker of the "Motor Neuron 2" pair of motor neurons (MN2) that form *en passant* synapses at sites of AChR clustering in the tail muscles [66]. Double ISH for *Nova* and *Islet* revealed co-expression in MN2 at ~stage 24 (15 hpf at 18 °C, Fig 2G). We confirmed the motor neuron identity of these posterior-most *Nova*-positive cells as motor neurons by performing ISH for *Nova* in embryos electroporated with *Fgf8/17/18>H2B::mCherry* plasmid (S5B Fig), which marks the A9.30 lineage of Ciona [64]. MN2 cells are derived from the A9.32 lineage and are invariantly positioned immediately posterior to the A9.30 lineage [67,68]. Indeed, we detected *Nova* expression in the cell just posterior to the A9.30 lineage, not co-expressed with H2B::mCherry, confirming its expression in MN2 (S5B Fig). Finally, ISH also revealed that *Agrin* is transcribed throughout the MG, in addition to other cells around the larval brain and sensory vesicle (Fig 2H). Taken together, these data show that *Nova* and *Agrin* are co-expressed in the larval motor neurons that form NMJs with the tail muscles.

## A minigene assay to study regulation of *Ciona Agrin* splicing in cell culture

Given their co-expression in *Ciona* neurons, we investigated whether Nova might also promote inclusion of the Z exons during alternative splicing of *Agrin* in *Ciona*, as Nova1/2 proteins do in vertebrates [43]. To do this, we developed a *Ciona Agrin* minigene splicing assay based on similar previously described assays [69–71]. We co-transfected plasmids encoding exons 40, 41, and the intervening introns and Z exons under the *cis*-regulatory control of the CMV promoter, together with different concentrations of *Ciona* or mouse Nova expression plasmids, into cultured mammalian cells (Fig 3A). We then assayed the inclusion of the Z exons in the resulting *Ciona Agrin* mini-transcripts by RT-PCR on cDNA prepared from transfected cells. Western blot and GFP fluorescence confirmed the expression of *Ciona* Nova proteins in mammalian cells (S6A and S7 Figs). As expected, inclusion of *Ciona Agrin* Z exons increased with higher doses of *Ciona* Nova (Fig 3B), which was observed even using different *Ciona* Nova isoforms (S6B Fig). Curiously, only Z11 and Z5 isoforms were detected by RT-PCR (Fig 3B) and confirmed by cloning and sequencing (see S1 File). This suggests that inclusion of the

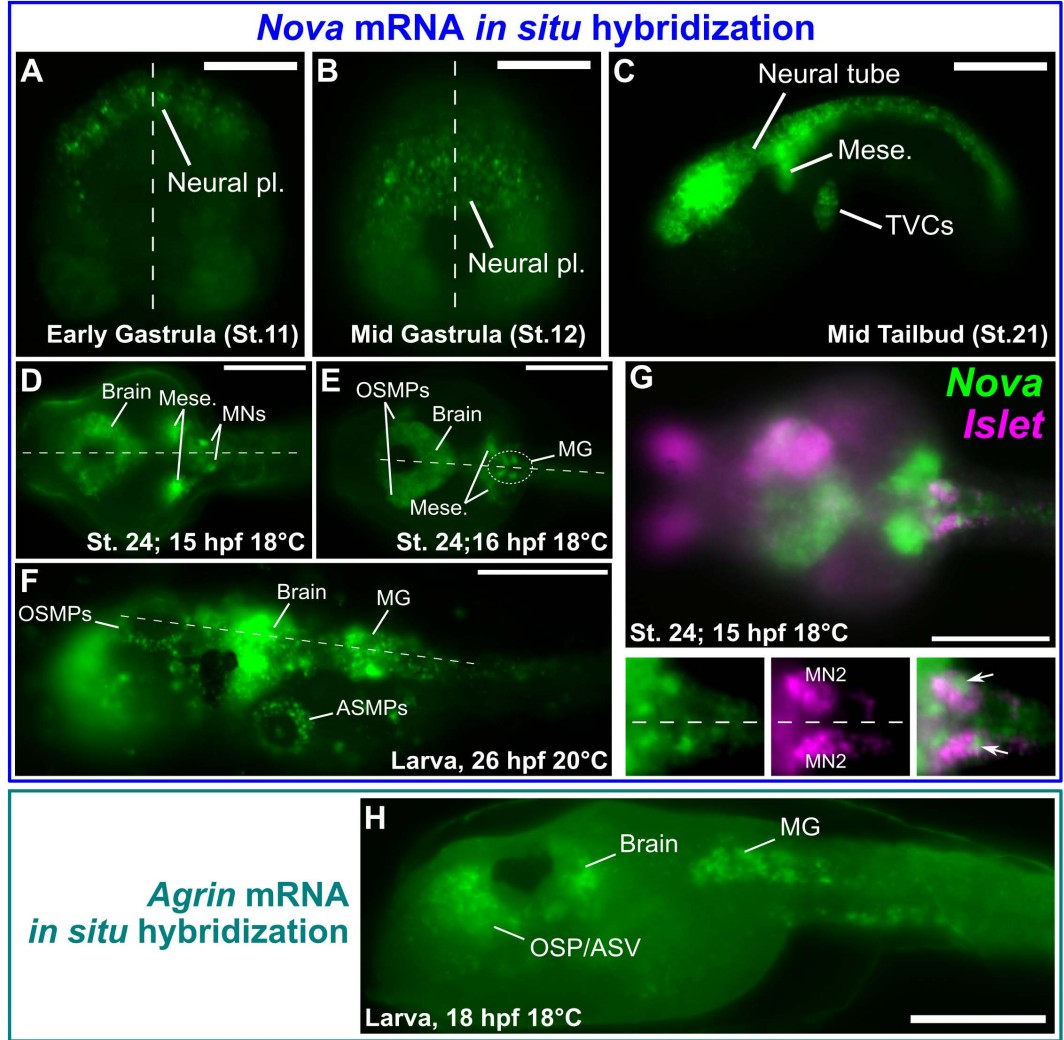

**Fig 2. Expression of *Nova* and *Agrin* determined by in situ hybridization. (A)** Whole-mount fluorescent mRNA in situ hybridization (ISH) of *Nova* in *C. robusta,* showing earliest detectable zygotic expression in neural plate progenitors at the early gastrula stage (Hotta Stage 11, approximately 4.5 h post-fertilization (hpf) at 20 °C. **(B)** Expression is observed in the nascent neural plate at Stage 12 (approximately 5 hpf at 20 °C). **(C)** *Nova* is expressed throughout the neural tube and also in mesoderm-derived mesenchyme and trunk ventral cells (TVCs, also known as cardiopharyngeal progenitors) at mid-tailbud stage, Stage 21 (approximately 9.5 hpf at 20 °C). **(D)** Expression of *Nova* is now observed to be maintained/upregulated in motor neurons of the Motor Ganglion (MG), as well as in the brain/posterior sensory vesicle and mesenchyme in earlier Stage 24 embryos (approximately 15 hpf at 18 °C). **(E)** *Nova* is also seen to be activated in oral siphon muscle progenitors (OSMPs) in later Stage 24 embryos (approximately 16 hpf at 18 °C). **(G)** Two-color double ISH of *Nova* (green) and *Islet* (magenta) showing co-expression in the bilaterally symmetric Motor Neuron 2 (MN2) pair of cells (arrows). Note that *Nova* mRNA distribution appears more nuclear than *Islet* in these cells at this stage. **(H)** ISH of *Agrin* in the *C. robusta* larva showing expression in the oral siphon/anterior sensory vesicle (OSP/ASV) region, in the larval brain, and in the Motor Ganglion (MG). Dashed lines in any panel indicate embryonic midline in dorsal views. All scale bars = 50 μm.

Z6 exon alone might be promoted by additional factors not present in our minigene assay. Strikingly, mouse Nova1 and Nova2 were unable to promote Z exon inclusion in the *Ciona Agrin* mini-transcripts (Figs 3B and S8A), even though they can still promote Z exon inclusion in mouse *Agrin* minigene transcripts (Fig 3C). This suggests that, although the regulation of *Agrin* splicing by Nova proteins might be conserved from tunicates to vertebrates, there may have been additional co-evolution that has resulted in divergent *cis/trans* compatibility: only *Ciona* Nova, not vertebrate Nova1/2, might be capable of promoting the inclusion of Z exons in *Ciona Agrin*.

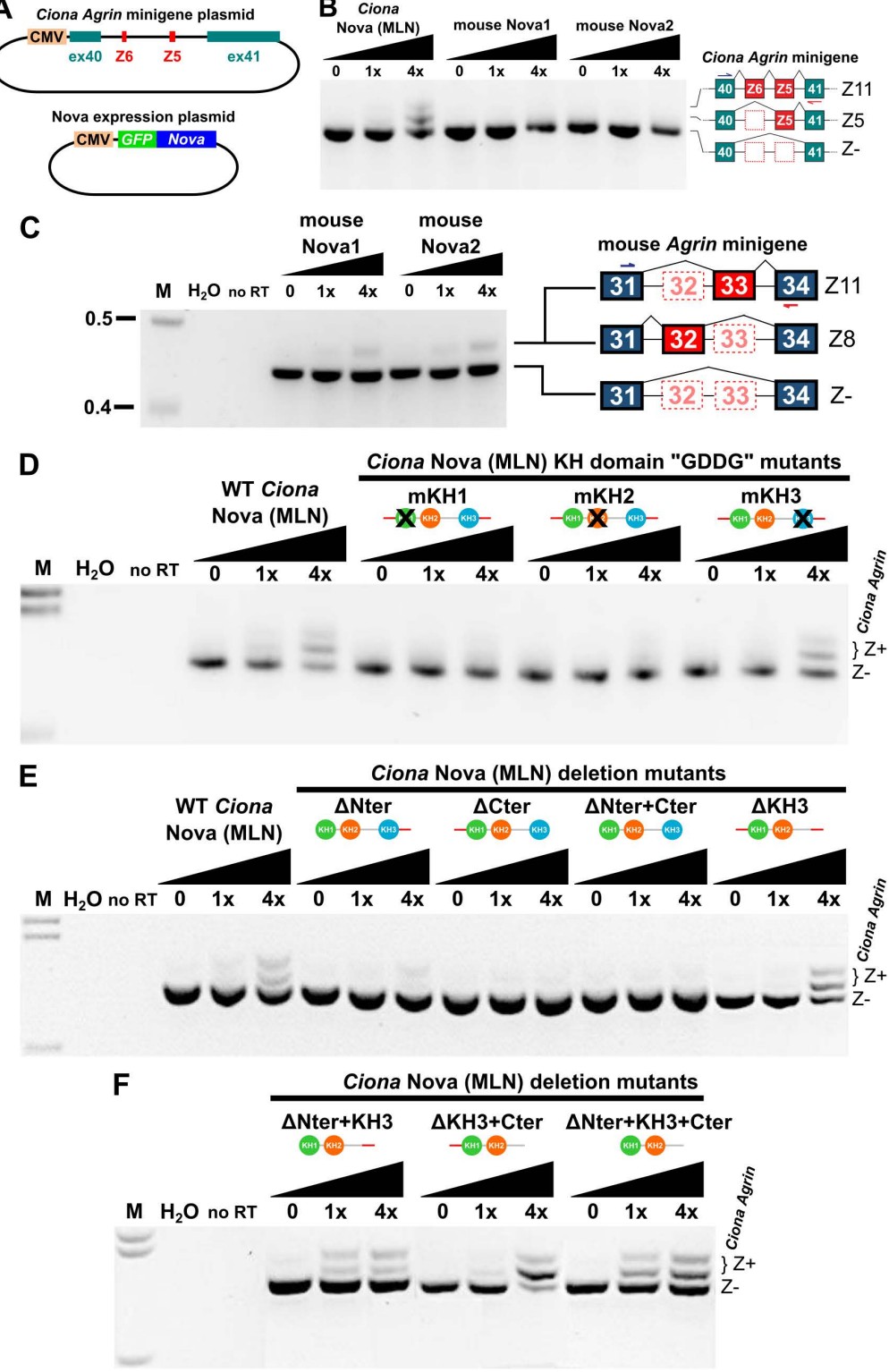

**Fig 3. *Ciona Agrin* minigene Z exon inclusion assay. (A)** Diagram of *Ciona robusta Agrin* minigene plasmid (top) used for alternative splicing assays in cultured mammalian (HEK293T) cells, along with Nova expression plasmids (bottom). **(B)** *Ciona* Nova ("MLN" isoform tested) can promote *Ciona Agrin* minigene Z exon inclusion, assayed by RT-PCR, while mouse Nova1 or Nova2 cannot. Identity of Z11, Z5, and Z⁻ (Z-negative) was confirmed

by cloning and sequencing RT-PCR products. Z6 isoforms were not detected in the minigene assay. Black slope indicates increasing Nova expression plasmid dose. **(C)** Mouse Nova1 and Nova2 can promote Z exon inclusion in a mouse *Agrin* minigene assay, confirmed by cloning and sequencing the RT-PCR product shown (see S1 File). **(D)** Testing the effect of "GDDG" mutations in each RNA-binding KH domain of *Ciona* Nova (KH1-3) using the same minigene assay as above. Abolishing the RNA-binding activity of KH1 and KH2, but not KH3, disrupts the ability of *Ciona* Nova to promote Z exon inclusion. **(E)** Deleting the N-terminus of *Ciona* Nova (MLN isoform) abolishes its ability to promote *Agrin* Z exon inclusion in the minigene assay. **(F)** This effect is nullified by concomitant deletion of the KH3 domain (see text for details). M: DNA molecular weight marker (in kilobase pairs). $H_2O$: using water instead of cDNA template for PCR. no RT: no reverse transcriptase added.

In vertebrates, both Nova1 and Nova2 can bind pre-mRNAs through their three KH domains, which are all conserved in *Ciona* Nova (Fig 1F). However, it is not currently known which KH domains in Nova mediate *Agrin* Z exon inclusion. Different KH domains of Nova1/2 can bind different RNA targets, resulting in complex mechanisms of binding and splicing by these proteins [44,72–75]. To test which KH domains of *Ciona* Nova are required for its ability to splice *Ciona Agrin* to include the Z exons, we used our minigene assay to test different KH domain mutants of the more widely-expressed "MLN" isoform of Nova. The three KH domains of *Ciona* Nova were disrupted (individually or in combination) by changing the G-X-X-G loop sequence to G-D-D-G, which impairs RNA binding without affecting domain stability [76]. According to our assay, we determined that the KH1 and KH2 domains of *Ciona* Nova are required for optimal Z exon inclusion, while disrupting the KH3 domain did not have any noticeable effect (Figs 3D, S8B and S8C). The same results were obtained with KH domain mutants of the "MMM" isoform as well (S9 Fig).

Surprisingly, deleting the N-terminus and/or C-terminus of *Ciona* Nova also abolished its ability to promote Z exon inclusion (Fig 3E). This effect was rescued by concurrently deleting the KH3 domain (Fig 3F), even though the KH3 deletion on its own did not affect Z exon inclusion (Fig 3E). These results were also obtained using equivalent deletion mutants of the "MMM" isoform (S9 Fig). Based on these data, we propose that the N- and C-termini of *Ciona* Nova comprise regulatory domains that prevent the KH3 domain from exerting an inhibitory effect on the KH1/KH2 domains that are essential for *Agrin* Z exon inclusion.

Finally, we asked whether there were any *cis*-regulatory sequences in the *Agrin* pre-mRNAs that might be important for its Nova-dependent splicing and Z exon inclusion. Indeed, we identified the presence of 18 potential Nova binding sites (YCAY) in the intron between exons Z5 and 41 (Fig 4A), with no other intronic YCAY sequences present elsewhere in this region. As vertebrate Nova proteins have been shown to bind pre-mRNAs via intronic YCAY clusters [73,77–79], we tested whether disrupting these sequences in the *Ciona Agrin* minigene plasmids might block the ability of *Ciona* Nova to promote Z exon inclusion. Indeed, we found that generating point mutations in some of these YCAY clusters suppressed the inclusion of *Ciona Agrin* Z exons by *Ciona* Nova (Figs 4B and S10). A footprint analysis indicated that the most crucial clusters mapped to YCAY sites 1–6 and 11–14 in the intron between the Z5 exon and constitutive exon 41 (Fig 4C). Therefore, *Ciona* Nova uses two Nova Intronic Splicing Enhancers, or NISEs [78], to promote Z exon inclusion, which requires at least two YCAY sequences in each element. Taken together, our minigene data indicate that *Ciona* Nova can promote the alternate splicing of *Agrin* pre-mRNAs through direct interactions between its KH1/KH2 domains and the intronic YCAY clusters in its target.

## A conserved Agrin-Lrp4 pathway for AChR clustering at the NMJ

In mammals, Z+ Agrin released by Nova-expressing MNs promotes AChR clustering in target muscles by binding to Lrp4 [23,24,43]. Thus, we sought to test the potentially conserved role of Nova in regulating *Agrin* alternative splicing and downstream neuromuscular synapse development in *Ciona.* To do this, we turned to tissue-specific CRISPR/Cas9 [80]. First, we aimed to establish the role of Z+ Agrin in controlling the clustering of AChRs in the larval tail muscles at *en passant* synapses formed by MN2 [66]. We designed eight different single-chain guide RNAs (sgRNAs) targeting sequences flanking the Z exons in *Ciona Agrin*, a region spanning exons 39–41. To test if CRISPR/Cas9 using these sgRNAs could

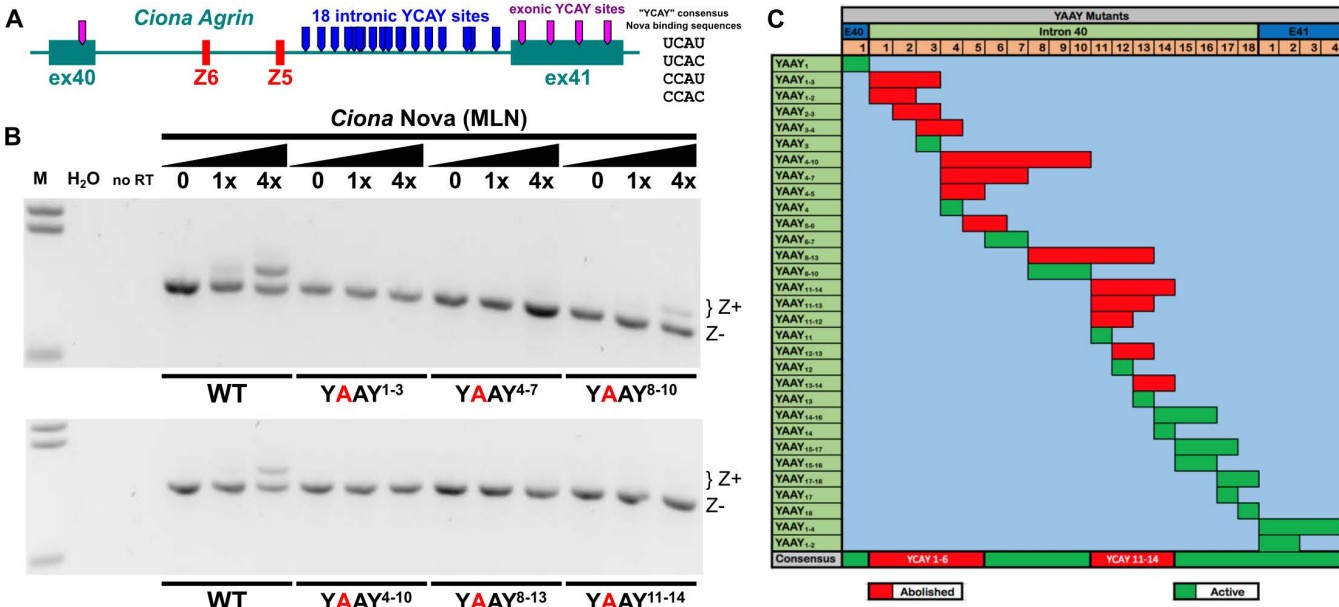

**Fig 4. Minigene assay to test *cis*-regulatory elements that promote Z exon inclusion. (A)** Diagram of the Z exon region of *Ciona robusta Agrin,* indicating 18 potential Nova binding sites (consensus: YCAY) in the intron between exon Z5 and constitutive exon 41, as well as exonic YCAY sequences (magenta tabs). **(B)** *Ciona Agrin* minigene Z exon inclusion detected by RT-PCR, using minigenes bearing different combination of candidate Nova-binding site mutations (YCAY > YAAY) predicted to disrupt Nova binding. Full set of mutations assayed shown in S10 Fig. M: molecular weight marker in kilobase pairs. $H_2O$: using water in place of cDNA template for PCR. no RT: no reverse transcriptase added. Equivalent bands on the right appear lower due to uneven migration during electrophoresis. **(C)** Chart summarizing effect of disrupting different YCAY sites in the *Ciona Agrin* minigene. Two clusters of intronic YCAY sites appear to be required for proper Nova-dependent Z exon inclusion: YCAY sites 1–6 and 11–14. Other intronic sites and sites in constitutive exons 40 or 41 are not required for Z exon inclusion.

suppress Z exon inclusion, we performed qPCR on cDNAs generated from embryos co-electroporated with *Eef1a>Cas9,* to drive ubiquitous Cas9 expression [81], together with different combinations of our *Agrin*-targeting sgRNA constructs. Three out of four such combinations resulted in reduced Z11+ *Agrin* amplification (Fig 5B). We selected the subset of sgRNAs that showed the highest and most specific reduction in Z11+ transcript levels (combination #1) for further investigation (Fig 5A).

We performed tissue-specific CRISPR/Cas9-mediated mutagenesis of the *Agrin* Z exon (*Agrin^{Z+}*) region by co-electroporating the selected sgRNA combination #1 (sgRNAs 1, 3, 5, 6, 7, and 8) together with *Sox1/2/3>Cas9* plasmid. The *Sox1/2/3* promoter was used to drive Cas9 expression in neural progenitors, including the lineage that gives rise to the motor neurons of the *Ciona* larva [81]. To assay AChR clustering at NMJs, we co-electroporated *Tbx6-r.b>AChRA1::GFP* plasmid to express in the tail muscles the AChRA1::GFP subunit fusion that was previously used to visualize such clusters postsynaptic to MN2 [66]. Neural-specific disruption of *Agrin^{Z+}* significantly reduced AChRA1::GFP clusters in the tail muscles, at dorsal sites of contact with MN2 (Figs 5C and S11). This effect was replicated using two different combinations of sgRNAs targeting more specifically the introns flanking the Z exons (sgRNAs 1 and 3), or exons 39 and 41 (sgRNAs 5 and 8) (Fig 5D), which were validated as generating indels at their appropriate target sites by genomic DNA amplicon sequencing (S12 Fig).

To test whether Lrp4 might play a conserved role as a receptor for Agrin in the tail muscles of *Ciona,* we performed muscle-specific CRISPR/Cas9-mediated disruption. To target the *Lrp4* gene specifically in muscles, we co-electroporated *Tbx6-r.b>Cas9* together with validated *Lrp4*-targeting sgRNAs (S13 Fig). Compared to the negative control, we found a significant reduction in the number of muscle cells with visible AChRA1::GFP clusters (Fig 5E and 5F). Often the

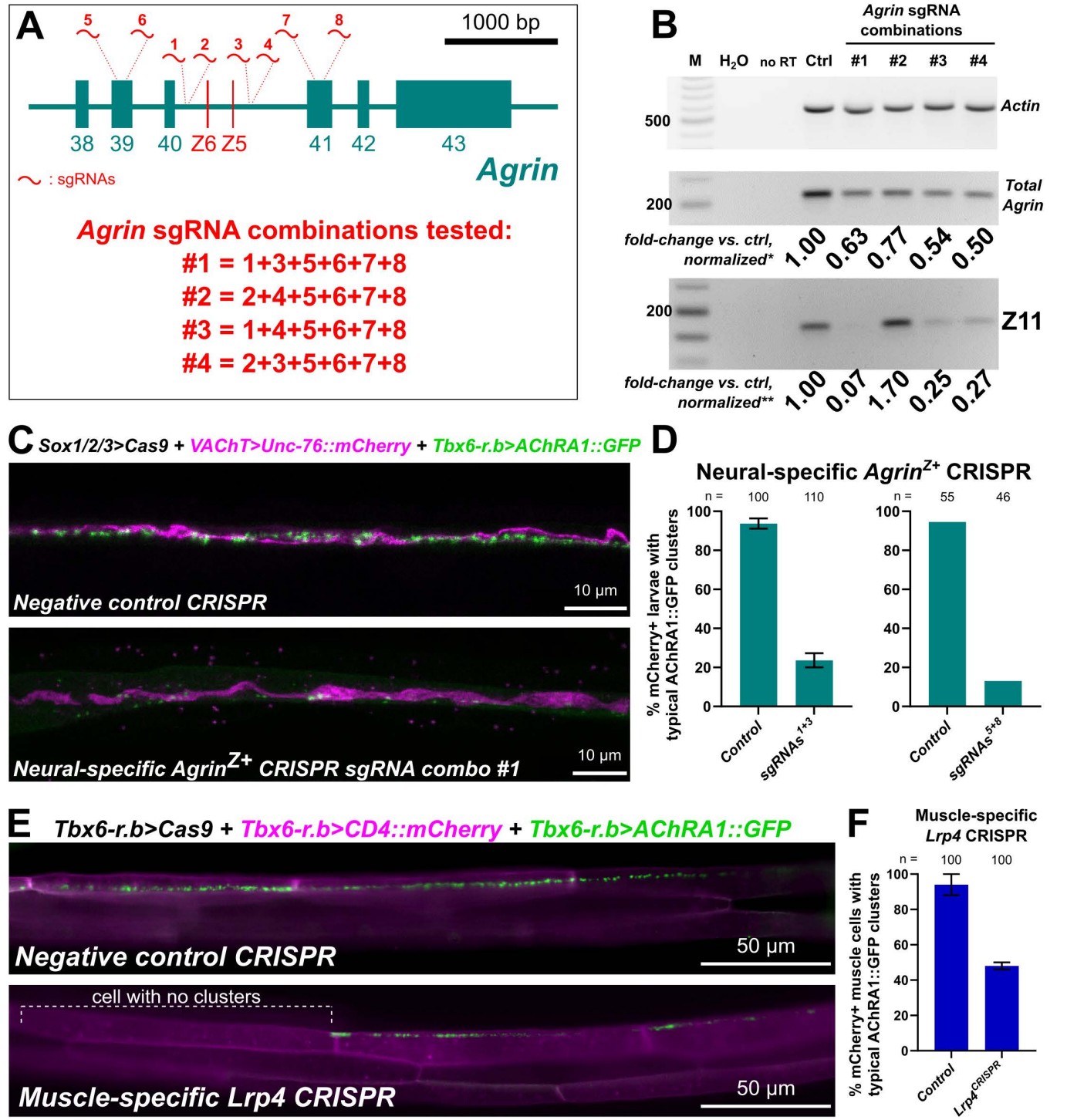

**Fig 5. Tissue-specific CRISPR/Cas9-mediated mutagenesis of *Agrin* and *Lrp4*.** (A) Partial diagram of the *Agrin* gene from *Ciona robusta,* showing position of target sites and combinations of sgRNAs for CRISPR/Cas9-mediated mutagenesis. (B) RT-PCR-based quantification of *Agrin* Z exon inclusion (from larvae collected at 22.5 h post-fertilization at 20 °C) following CRISPR-mediated disruption of sequences surrounding the Z exons. Selected sgRNA combinations are indicated in panel **A**, which were co-electroporated with *Eef1a>Cas9*. Fold-change of band intensity is compared to a negative control CRISPR sample. * total *Agrin* bands were normalized according to corresponding *Actin* band intensity. ** Z11 exon bands were normalized according to both *Actin* and total *Agrin* bands for each corresponding sample. M: molecular weight marker in base pairs H$_2$O: water used in place of

cDNA template for PCR. no RT: no reverse transcriptase added. **(C)** Neural-specific CRISPR-mediated disruption of *Agrin* using sgRNAs indicated in the diagram above results in loss of Acetylcholine receptor A1::GFP clusters (AChRA1::GFP, green) in tail muscles, driven by the *Tbx6-r.b* promoter [98]. Motor neuron axons labeled by *VAChT > Unc-76::mCherry* in magenta [99]. **(D)** Scoring loss of AChRA1::GFP clusters in the muscles upon targeting *Agrin* using more specific combinations of NGS-validated sgRNA pairs (1 + 3 and 5 + 8). 1 + 3 sgRNA pair tested in duplicate with at least 45 larvae in per condition and duplicate. Negative control larvae electroporated with negative control sgRNA instead. **(E)** Muscle-specific CRISPR-mediated disruption (using *Tbx6-r.b > Cas9*) of the Agrin receptor-encoding gene *Lrp4* shows similar loss of AChRA1::GFP clusters. Effects were seen on a muscle cell-by-cell basis, as expected if the effect of disrupting the receptor is cell-autonomous. Negative control is actually muscle-specific CRISPR-mediated disruption of *Nova,* confirming neural-specific requirement of Nova as demonstrated further below in Fig 6 **(F)** Scoring of larvae represented in the previous panel. Experiment performed in duplicate with 50 individual muscle cells examined per condition and duplicate. Data underlying panels **D and F** can be found in S1 Data file.

AChRA1::GFP clusters were either present or entirely absent from whole muscle cells (Fig 5E), which in *Ciona* larvae are invariantly derived from different Tbx6-r.b-positive precursors and therefore likely experience independent CRISPR/Cas9 mutagenesis events due to mosaic uptake of electroporated plasmids [82]. Taken together, these data suggest that, similar to its role in vertebrates, Lrp4 is also required for AChR clustering in *Ciona* NMJs.

### *Ciona* Nova is required for *Agrin* Z exon inclusion and AChR clustering

We next asked whether in *Ciona* Nova plays a conserved role in splicing of neural-specific, Z⁺ *Agrin* mRNAs to induce AChR clustering at the NMJ. First, we designed and validated three sgRNAs targeting the *Nova* gene by CRISPR/Cas9 (Figs 6A and S14). We then used RT-PCR to investigate *Agrin* Z exon inclusion upon CRISPR/Cas9-mediated disruption of *Nova* in *Ciona* larvae (Fig 6B). Z⁺ *Agrin* transcripts were significantly reduced upon co-electroporation of *Eef1a > Cas9* and any one of the three *Nova*-targeting sgRNAs individually, or in combination (Fig 6B and 6C). This effect was reproduced in triplicate, while overexpression of Nova (*Eef1a > Nova*) resulted in increased Z exon inclusion (Fig 6C). Taken together, these results suggest that *Ciona* Nova is necessary and sufficient for *Agrin* Z exon inclusion *in vivo*, just like Nova1/2 in mammals [43]. When we looked at AChRA1::GFP clusters at NMJs, we saw that they were reduced in frequency or density in *Nova* CRISPR larvae compared to negative control larvae (Fig 6D–6F). Furthermore, AChR clustering was partially rescued by expressing CRISPR-insensitive *Nova* cDNA in MN2 (Fig 6E). Taken together, these results reveal that a Nova-Agrin-Lrp4 pathway for AChR receptor clustering at the neuromuscular synapse is conserved between mammals and *Ciona*.

### Expression of *Nova* in neurons is activated by the transcription factor Ebf

Although the role of Nova in regulating neural *Agrin* isoform splicing has been established [43], almost nothing is known about how *Nova* itself is activated in motor neurons. To help understand the transcriptional regulation of *Ciona Nova*, we isolated approximately 2 kbp sequence immediately 5′ to exon 1b of *Nova,* cloning it upstream of GFP (*Nova[1b] − 2011/ + 6 > GFP,* or simply *Nova > GFP*) (Fig 7A). This drove strong GFP expression in MG neurons, in addition to some brain neurons, the otolith, and oral siphon muscle precursors (Fig 7B), recapitulating much of the expression observed by ISH (see Fig 2). A smaller fragment upstream of exon 1a *(Nova[1a] − 922/-1 > GFP)* did not drive noticeable expression outside the mesenchyme and some epidermal cells (S15 Fig).

One of the major sequence-specific transcription factors expressed in the differentiating neurons of the *Ciona* MG is Ebf [83], the sole *Ciona* ortholog of EBF family factors in vertebrates, also known as COE (Collier/Olf/EBF) [84]. In *Ciona*, Ebf is expressed in differentiating MG neurons and is required for MN2 specification [52,81], while in vertebrates Ebf2 is required specifically for axial MN development [85]. We therefore sought to investigate the role of Ebf in regulating *Nova* in the *Ciona* MG.

To test if *Ebf* is required for *Nova* expression in differentiating MG neurons in *Ciona*, we targeted it for neural tissue-specific CRISPR/Cas9-mediated disruption using a previously published, highly efficient sgRNA [86]. We assayed loss of

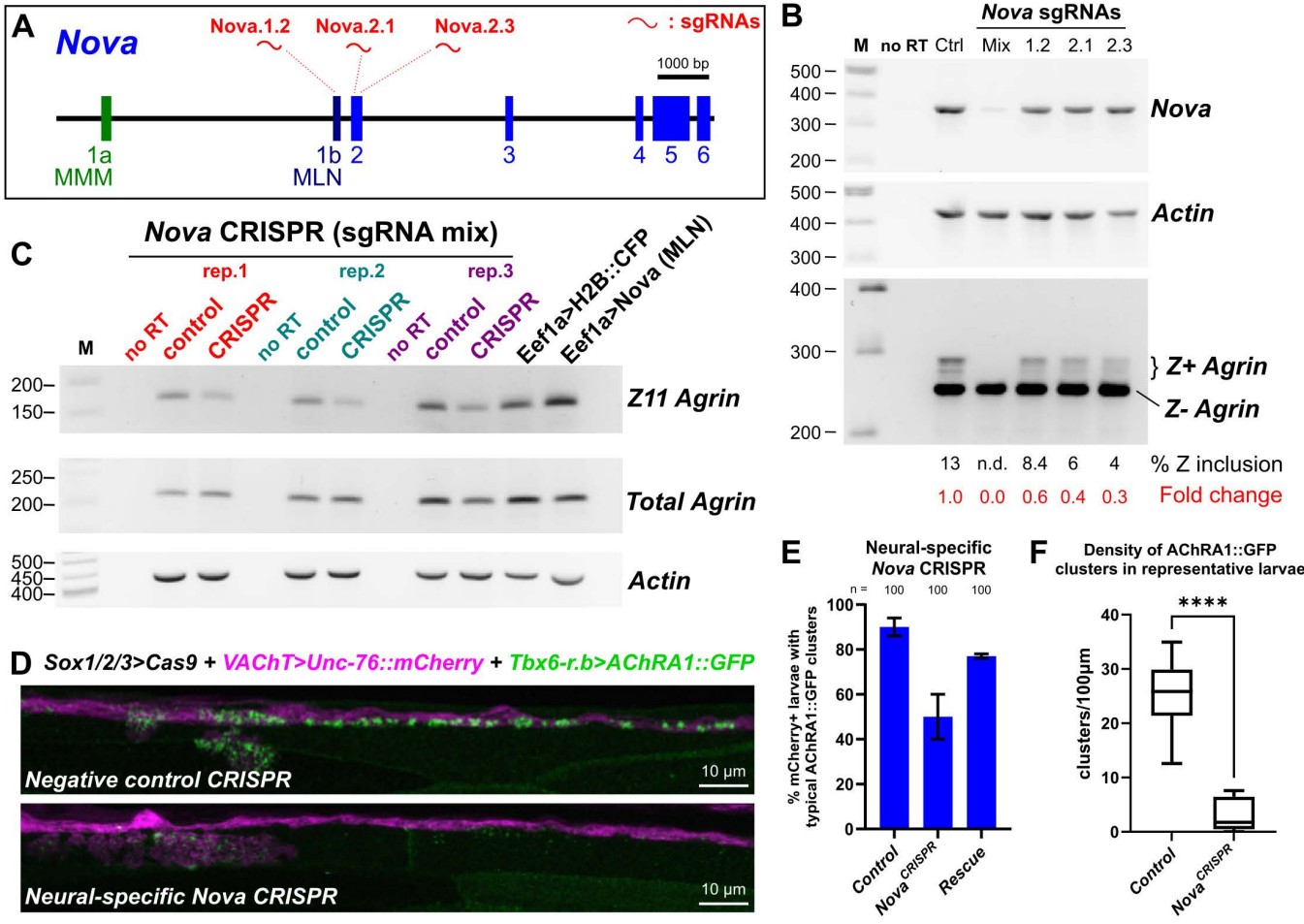

**Fig 6. Neural-specific disruption of *Nova* greatly reduced AChR clustering in muscles. (A)** Diagram of the *Nova* gene in *Ciona robusta,* showing target sites of the three selected *Nova*-targeting sgRNAs. **(B)** RT-PCR assay measuring reduction of *Agrin* Z exon inclusion in larvae upon CRISPR-mediated mutagenesis of *Nova*, compared to a negative CRISPR control sample ("Ctrl"). All three sgRNAs resulted in some reduction of Z exon inclusion on their own, when co-electroporated with neural-specific *Sox1/2/3 > Cas9*, but the largest reduction was seen when all three sgRNAs were combined ("Mix") and co-electroporated with ubiquitously activated *Eef1a > Cas9*. Only the mix substantially reduced *Nova* transcript detection, perhaps due to high frequency of large deletions spanning the target amplicon. M: molecular weight marker in base pairs. no RT: no reverse transcriptase added. **(C)** All three replicates of ubiquitous *Nova* CRISPR (using *Eef1a > Cas9* and all three sgRNAs combined) show reduction of *Agrin* Z exon band, reproducing the effects seen in panel **B**. A slight increase in Z exon inclusion is seen upon Nova (MLN isoform) overexpression using the *Eef1a* promoter. **(D)** Neural-specific CRISPR/Cas9-mediated disruption results in decreased AChRA1::GFP clustering in tail muscles, phenocopying neural-specific, CRISPR-mediated disruption of *Agrin* Z exons. Negative control CRISPR performed using *U6 > Control* negative control sgRNA plasmid. **(E)** Scoring of larvae represented by the panel above. The loss of AChRA1::GFP clustering was rescued by co-electroporation of a CRISPR-insensitive *Islet − 7,216/−3,950 + bpFOG > Nova(MLN)* rescue plasmid, demonstrating specificity of the CRISPR effect. Fifty larvae examined per duplicate and condition. **(F)** Local densities of AChRA1::GFP clusters were quantified in 10 representative larvae in either negative control or *Nova* CRISPR condition (as in panel **D**) using confocal imaging. **** indicated $p < 0.0001$ following an unpaired *T* test using Welch's correction. Data underlying panels **E and F** can be found in S1 Data file.

*Nova > GFP* reporter gene expression in the MG neurons of *Ebf* CRISPR larvae compared to control larvae. Neural-specific disruption of *Ebf* (using again *Sox1/2/3 > Cas9*) resulted in significant, near total loss of *Nova > GFP* in MG neurons (Fig 7C and 7D). Taken together, these data suggest that Ebf is necessary for neuron-specific expression of *Nova* in *Ciona* embryos. Of note, *Nova > GFP* was not lost from the otolith, which does not express *Ebf*, suggesting that *Nova* is regulated independently of Ebf in this cell.

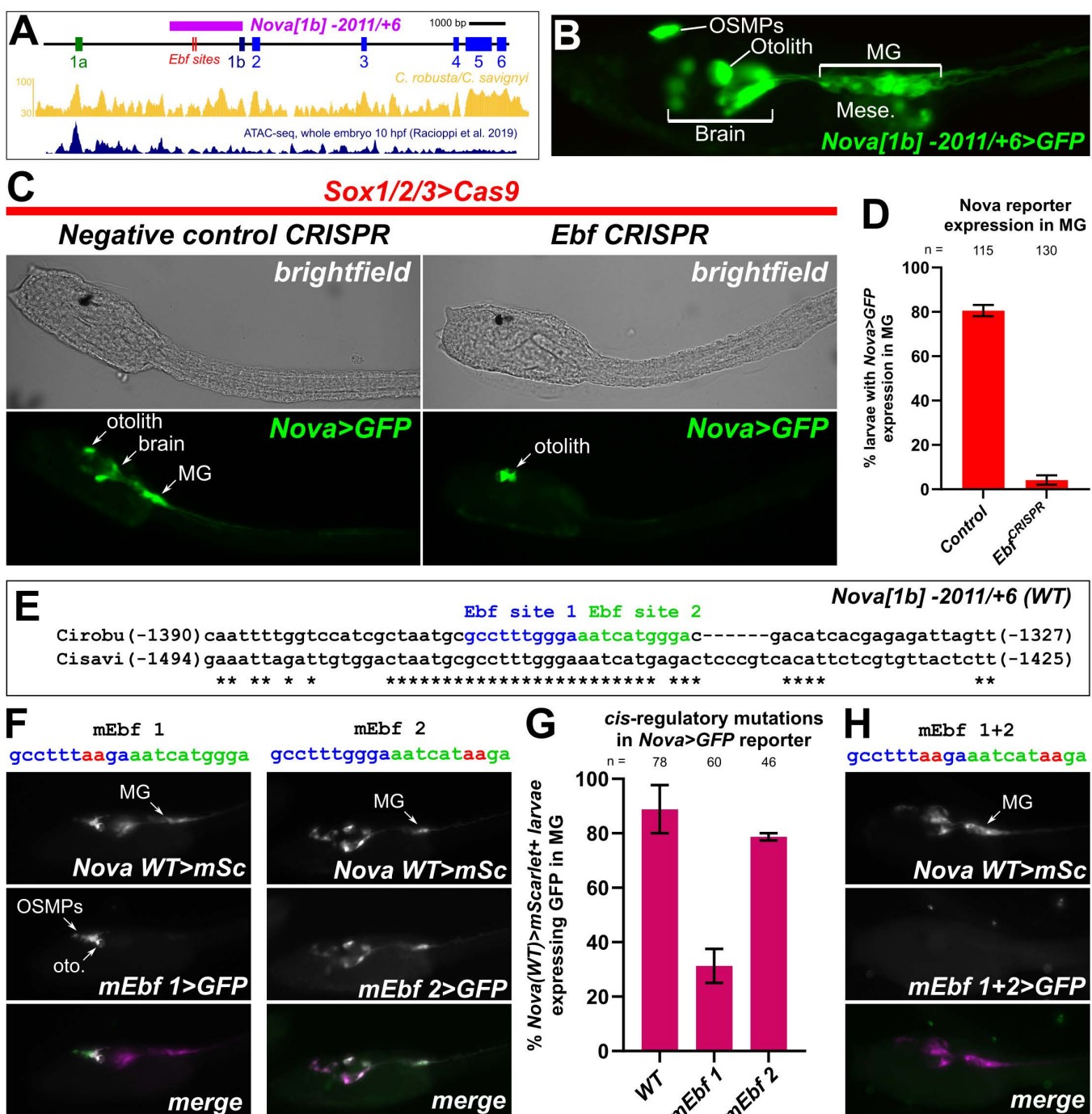

**Fig 7. Ebf regulates transcription of *Nova* in *Ciona* motor neurons.** (A) Diagram of *Nova* locus in *Ciona robusta,* showing position of the cloned *Nova[1b] − 2011/ +6* promoter. Conservation with the *C. savignyi* genome shown as golden peaks below, as visualized in the ANISEED database [100]. ATAC-seq peaks from Racioppi and colleagues [101] shown below, predicted Ebf binding sites indicated as red lines. (B) *C. robusta* larva electroporated with the *Nova[1b] − 2011/ +6>GFP* reporter plasmid, recapitulating expression seen by in situ hybridization in the motor ganglion (MG), larval brain/ sensory vesicle (including the otolith pigment cell), and oral siphon muscle progenitors (OSMPs). (C) Neural-specific CRISPR-mediated disruption of *Ebf* eliminates expression of the *Nova* reporter plasmid in all cells except in the otolith. Negative control performed using *U6>Control* negative control sgRNA instead. (D) Scoring of *Nova* reporter expression (as represented in previous panel) in duplicate, with at least 50 larvae examined per duplicate and condition. (E) Sequence alignment between *C. robusta* and *C. savignyi* genomic sequences showing conserved, predicted Ebf binding sites approximately 1.4 kb upstream of *Nova* exon 1b. (F) Effects of *C. robusta Nova* GFP reporter plasmid (green) bearing targeted mutations predicted to disrupt

Ebf binding to Ebf sites 1 and 2 (mEbf 1 and mEbf 2, respectively), by co-electroporating the wildtype *Nova* mScarlet (mSc) reporter plasmid (magenta). **(G)** Scoring of larvae represented in previous panel, showing more substantial effect of disrupting predicted Ebf site 1 than Ebf site 2. Electroporations performed and assayed in duplicate, with sample size of at least 15 larvae examined per duplicate per construct. **(H)** Representative image showing complete loss of *Nova* reporter expression upon mutating both predicted Ebf sites. Data underlying panels **D and G** can be found in S1 Data file.

To get a better understanding of whether *Nova* is a direct or indirect transcriptional target of Ebf, we searched the *Nova* 5′ *cis*-regulatory sequence for potential Ebf binding sites. The predictive algorithm JASPAR only found two such sites, in tandem approximately 1,300 bp upstream of the transcription start site and almost perfectly conserved in the related species *C. savignyi* (Fig 7E). When we mutated the first Ebf site (mEbf 1), reporter expression was greatly reduced in MG and brain neurons, but not in the oral siphon muscle precursors or otolith (Fig 7F and 7G). In contrast, when we mutated the second Ebf site (mEbf 2), reporter expression was not significantly reduced (Fig 7F and 7G). However, when both sites were mutated in the same construct, we lost all expression in Ebf-positive cells (Fig 7H), suggesting that the Ebf 2 site might serve as a "backup" site for the primary site, Ebf 1. Taken together, we conclude that Ebf is a key activator of *Nova* transcription during motor neuron differentiation in *Ciona*.

## Discussion

Here we have shown that a conserved alternative splicing-based switch for AChR clustering at neuromuscular synapses evolved before the evolutionary divergence of tunicates and vertebrates (Fig 8A). In this conserved pathway, the RNA-binding protein Nova is expressed in motor neurons and promotes the inclusion of *Agrin* "Z" microexons, which encode a short peptide motif that mediates the activation of Lrp4 and downstream clustering of AChRs post-synaptically in muscle cells. A potential Z exon encoding the key N-X-I/V motif needed for interaction with Lrp4 was also identified in

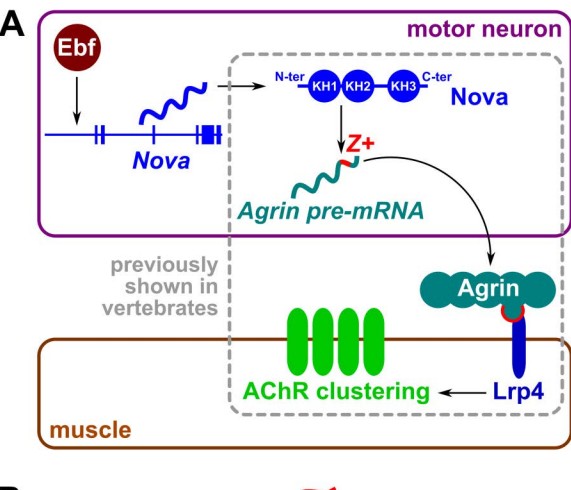

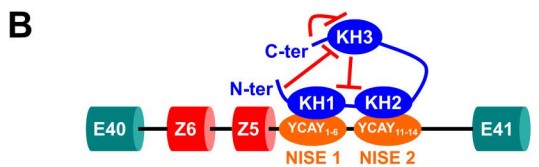

**Fig 8. Summary model diagram of a conserved *Nova-Agrin-Lrp4* pathway for AChR clustering in *Ciona*. (A)** Summary of conserved pathway, and previously unidentified Ebf-Nova regulatory connection identified in *Ciona*. **(B)** Model of proposed mechanism for *Ciona* Nova-dependent alternative *Agrin* splicing via binding of KH1 and KH2 domains to two YCAY-rich NISEs identified in the intron 3′ to the Z exons (though not known in what exact orientation or configuration). Inhibitory effect of KH3 domain is relieved by the N- and C-termini. Diagram not drawn to scale. See text for details.

an amphioxus *Agrin* gene (S2 Fig), pushing the evolutionary emergence of this mechanism at least as far back as the last common chordate ancestor [53]. The expression and function of other potentially conserved components of this pathway downstream of Lrp4 (e.g., Musk, Dok-7) in non-vertebrate chordates remain to be fully investigated, although *Rapsyn* has been shown to be specifically expressed in developing larval tail muscles in *Ciona* [66].

Although we did not find potential Z exons in the *Agrin* genes of other invertebrates such as the nematode *C. elegans* or sea urchins (S3 Fig)*,* we noticed a highly conserved N-X-I/V motif in all vertebrate and invertebrate Agrin sequences, encoded by the constitutive exon just before the Z exons (S2 Fig). To our knowledge, this additional N-X-I/V motif has not been studied previously, and we do not know its function. It has been shown that N-X-I/V motifs can have either stimulatory or inhibitory effects on Lrp-dependent Wnt signaling, depending on the exact receptor or antagonist [87]. Therefore, it is possible that this ancestral, constitutive N-X-I/V motif might instead have an antagonistic effect on Lrp4, in the absence of Nova expression and inclusion of an additional, Z exon-encoded N-X-I/V motif.

It was not previously known which KH domains of Nova1/2 mediate Z exon inclusion, nor which *cis*-acting sequences might mediate Nova1/2 binding to *Agrin* mRNAs. Using a minigene assay, we found that *Ciona* Nova needs both KH1 and KH2 domains to promote *Ciona Agrin* Z exon inclusion, and this depends on two separate Nova-binding intronic splicing enhancer elements (NISEs). We propose a model where KH1 binds to the first NISE and KH2 to the second NISE, or vice-versa. We also discovered that at least two, and any two, consecutive YCAY sequences from either NISE1 or NISE2 are needed to promote Z exons inclusion (Fig 4), thus contributing to deciphering the splicing code that mediates Nova-dependent alternative splicing regulation.

We also revealed a potential autoinhibitory mechanism involving the N- and C-termini and the KH3 domain of *Ciona* Nova. We speculate that either terminus may act as a regulatory domain, either alone or in combination, that suppresses the inhibitory effect of KH3 on splicing activity. Thus, when the N- or C-terminus is deleted, KH3 exerts its inhibitory action and splicing is suppressed. However, when KH3 is deleted alongside N- or C-terminus, splicing activity is restored. One hypothesis is that this constitutes a mechanism to regulate whether splicing is happening via KH1 and KH2 – as in the case of *Agrin* Z exons – or via KH3, where KH3 may be available for interactions with pre-mRNAs encoded by other genes unless inhibited by the N- or C-terminal domains (Fig 8B). In such a model, Nova operates via two splicing modes: one mode acts via KH1 and KH2 to splice a subset of targets that includes *Agrin*, and the second mode uses KH3 to splice a different set of Nova targets. The N- and C-termini domain acts as a switch between the two splicing modes.

An alternative hypothesis is that the N-/C-termini regulate the negative effect of KH3 on the expression and/or stability of Nova. In fact, deleting either terminus appeared to have a negative effect on GFP::Nova fluorescence in transfected mammalian cells, which was seemingly rescued by concurrent deletion of KH3. The overall picture is likely to be more complex, and it may require further biochemical, crystallographic, and/or bioinformatical approaches to understand how different domains of Nova modulate its splicing activity. Nevertheless, our findings uncover previously unsuspected layers of regulation of Nova function.

Our results also unexpectedly suggested that the exact mechanisms of *Agrin* splicing by Nova might be divergent between tunicates and vertebrates, as mouse Nova1/2 proteins were unable to promote Z exon inclusion in a *Ciona Agrin* minigene assay. This is a stark example of Developmental System Drift, which refers to the evolutionary divergence of molecular mechanisms underlying otherwise conserved biological traits or phenomena [88,89]. Co-evolution between *Agrin* and *Nova* genes along the tunicate and/or vertebrate branches of the chordate tree might have resulted in this molecular "incompatibility" across species, even though Nova promotes *Agrin* Z exon inclusion in both groups. It remains to be seen whether the N-/C-terminus-dependent autoinhibitory mechanism that we identified is also deeply conserved, or whether its divergence can explain the apparent differences in Nova splicing mechanisms between tunicates and vertebrates. Further work in *Ciona*, amphioxus, and vertebrates will be required to investigate in greater detail the evolution of Nova structure-function in chordates.

 PLOS Biology

Finally, we have also shown that the transcription factor Ebf, an important terminal selector of motor neuron fate [52], is required to activate transcription of *Ciona Nova* in the motor ganglion. It was not clear if the loss of Ebf has a selective effect on *Nova* expression or a more general negative effect on neuronal specification. Ebf orthologs are also expressed in the vertebrate spinal cord, but appear to have selective effects on the differentiation of specific subsets of motor neurons [85]. This suggests the possibility that the regulatory connection between Ebf and *Nova* might also be conserved. This would in turn connect a largely RNA- and protein-based pathway for AChR clustering (Nova-Agrin-Lrp4) back to a transcriptional gene regulatory network downstream of motor neuron specification. However, future work will be needed to more precisely place the transcriptional regulation of *Nova* within overarching regulatory networks for chordate motor neuron differentiation.

## Materials and methods

### *Ciona* handling, electroporation, and transcriptome analysis

Adult *Ciona robusta* (*intestinalis Type A*) specimens were collected and shipped by M-REP (San Diego, California) and kept in artificial sea water tanks until use. Gametes were isolated and dechorionated as previously described [90]. Electroporations were performed as previously described [91], using plasmid DNA mixes defined in the S1 File. AChRA1::GFP plasmid [66] was kindly provided as a gift by Dr. Atsuo Nishino. For direct visualization of GFP/mCherry fluorescence, embryos and larvae were fixed in MEM-Formaldehyde buffer as previously described [92]. Fluorescence whole-mount mRNA in situ hybridizations were performed as previously described [93]. Embryos and larvae were imaged using upright or inverted epifluorescence or scanning-point confocal microscopes. Single-cell RNAseq plots were generated using custom web-based interfaces and published data [59], based on the ShinyCell R package [94].

### CRISPR/Cas9 methods in *Ciona*

Internet-based prediction algorithm CRISPOR [95](http://crispor.tefor.net/) was used to identify candidate sgRNAs for CRISPR/Cas9. Expression plasmids for sgRNAs were constructed by ligating annealed oligonucleotides [81], Gibson assembly of PCR products [80], or synthesized and custom-cloned *de novo* (Twist Bioscience, California). Validation of sgRNA efficacies was performed using either Sanger sequencing of amplicons following the "peakshift" method [80], or by Illumina-based next-generation amplicon sequencing [96]. All promoter, sgRNA, and primer sequences are listed in the S1 File.

### Minigene assay

The day before the transfection, $0.6 \times 10^6$ HEK293T cells were seeded per well in a 6-well plate (USA Scientific) in DMEM culture medium. On the day of transfection, a total of 2.5 μg DNA of minigene, cDNA construct, and empty vector was used to transfect each of 6 well plate(s) and 7.5 μL of linear polyethylenimine (PEI; Polysciences), MW 25,000 (1 mg/mL) was used in a ratio of 1:3 (DNA: PEI). A 0.5 μg (=1×) of minigene DNA was used in each well to test splicing with different amounts of splicing factor (0 μg = 0×, 0.5 μg = 1×, and 2.0 μg = 4×). Empty vector was used to bring the total amount of DNA to 2.5 μg (2.0 μg = 4×, 1.5 μg = 3×, and 0 μg = 0×) per well. The total volume of the DNA mixture was 200 μL. First, the exact amount of DNA in μL was pipetted in a 1.5 μL Eppendorf tube (Eppendorf) and Opti-MEM media (Thermo Scientific) was used to bring the volume to 192.5 μL. Then the mixture was vortexed thoroughly. Finally, 7.5 μL of PEI was added, vortexed, and centrifuged briefly. The mixture was then incubated for 15 min at room temperature. In the meantime, medium in the cells was aspirated and 2 mL of fresh DMEM medium was added. After a 15 min incubation, 200 μL of reaction mixture was added to the cells and the plate was cross-shaken gently. The plate was then incubated for 48 h at 37 °C. All cloning primers are listed in the S1 File. Mouse *Agrin* minigene plasmid was previously published [97] and shared as a kind gift by Dr. Robert B. Darnell.

## RT-PCR and qPCR

RNA from homogenized HEK293T cells was extracted 48 h after transfection using RiboZol RNA Extraction Reagent (AMRESCO) or IBI Isolate (IBI Scientific) according to the manufacturer's instructions. A total of 5 µg of RNA per sample was digested in a 50 µL reaction containing 1.5 µL of TurboDNase (Thermo Scientific), 5 µL of 10× Buffer, and double-distilled water (ddH$_2$O) to 50 µL. After 30 min of incubation at 37 °C another 1.5 µL of TurboDNase was added to the mixture and incubated for another 30 min. After a total of one-hour incubation 10 µL of TurboDNAse Inactivation Reagent was added and samples were kept at room temperature for 5 min, and the tubes were flicked every 2 min to resuspend the inactivation reagent. Then the tubes were centrifuged at 10,000 rpm for 90 s to collect supernatant for processing.

From total RNA we synthesized cDNA using RevertAid First Strand cDNA Synthesis Kit (Thermo Scientific). A mix of 250 ng RNA and 1 µL of oligo (dT)$_{18}$ at 500 ng/µL was prepared in a total volume of 12 µL (diluted in sterile ddH$_2$O). The mix was incubated for 5 min at 65 °C in a PCR machine. After this, an RT reaction mix was prepared combining the mixture above with the following ingredients in a total volume of 20 µL: 5× RT Buffer (4 µL); RiboLock RNase Inhibitor 20 U/µl (0.5 µL); 10 mM dNTPs (2 µL); RevertAid RT 200 U/µl (0.5 µl). This mixture was incubated for 1 h at 42 °C followed by 5 min at 72 °C in a PCR machine. After the incubation, 5 µL of water were added to each tube bringing the volume to a total of 25 µL. Each RT reaction mix had a concentration of 10 ng of starting RNA/µL. A 5 µL from each RT reaction, equivalent to 50 ng of starting RNA, were used as template for each RT-PCR.

A mixture of 10× PCR Buffer (5 µL); dNTPs 10 mM (1 µL); forward and reverse primers each 10 µM (1 µL); 5 U/µL HotStarTaq Plus DNA polymerase (Qiagen) (0.4 µL) or 5 U/µL Dream Taq Hot Start DNA polymerase (Thermo Scientific) (0.4 µL) and RT reaction (5 µL) in a total volume of 50 µL (diluted in sterile ddH$_2$O) was prepared in a PCR tube. The PCR reaction was performed with initial denaturation for 5 min at 95 °C; variable number of cycles of: denaturation for 30 s at 94 °C, annealing for 30 s at 60 °C and elongation for 30 s at 72 °C. This was followed by a final extension of 7 min at 72 °C and a hold at 12 °C. All primer sequences can be found in the S1 File.

## Western blot

Forty-eight hours after transfection, HEK293T cells were collected and resuspended in lysis buffer: 0.5% deoxycholic acid sodium salt, 0.1% SDS, 0.5% NP-40, 1× PBS and 50% glycerol with protease inhibitor cocktail (AMRESCO). The lysate was left on ice for 20 min. After sonication at 50% amplitude (Fisher Scientific Sonic Dismembrator Ultrasonic Processor FB120, 120 W 20 kHz), the lysate was subjected to centrifugation at 14,000 rpm at 4 °C for 15 min and the supernatant collected in a fresh tube. Total protein amounts were calculated using a standard Bradford assay and 10 µg of protein extract from each sample were loaded on a 8% SDS-polyacrylamide gel. Proteins were then transferred from gel onto PVDF membrane (Millipore, Immobilon-FL) and blocked in Odyssey blocking buffer (TBS, LI-COR). Membranes were blotted with mouse monoclonal anti-GFP antibody (Santa Cruz Biotech, sc-9996, 1:1000 dilution) and then donkey anti-mouse IRDye 800CW secondary antibody (LI-COR, 926–32,212, 1:5000 dilution). The signal was detected using the Odyssey CLx imaging system (LI-COR).

## GFP::Nova imaging in transfected mammalian cells

500,000 HEK293T cells were seeded per well in a 6-well plate. The following day cells were transfected with 500 ng of *Ciona Agrin* minigene and 2 µg of EGFP::Nova construct, following the transfection protocol used for the minigene splicing assays using 4× amount of splicing factor. After 48 h of transfection, cells were treated with Hoechst 33342 (Thermo Scientific) for 30 min according to the manufacturer. Fluorescence images of living cells were collected using an Axio Vert.A1 microscope (Zeiss) with an Infinity 2 CCD monochrome camera and Infinity Analyze software (Lumenera).

## Structural predictions using AlphaFold

Rat Agrin (Z8) and Lrp4 sequence fragments that were structurally analyzed previously [55] were identified and aligned to *C. robusta* Agrin (Z6) and Lrp4 to identify the corresponding sequences (see S1 File). Rat and *C. robusta* sequences were then entered into AlphaFold Server (alphafoldserver.com), the predicted structures were downloaded and the top predictions visualized further using the PyMOL Molecular Graphics System, Version 3.1 (Schrödinger, LLC). AlphaFold Server prediction files can be found here: https://osf.io/53sb6/.

## Supporting information

**S1 Fig. (A) Left: AlphaFold prediction suggesting the N-X-F motif of the *C. robusta* Agrin Z6 loop (bottom panel) might bind to the conserved residues in Lrp4 (see panel B), similar to what was previously determined for N-X-I in rat Agrin by X-ray crystallography (Zong and colleagues [55]).** Right: AlphaFold prediction of rat Agrin-Lrp4 interaction. **(B)** Alignment of rat and *C. robusta* ("Cirobu") Lrp4 sequences highlighting the residues important for N-X-I/V/F binding as color-coded in the ribbon diagrams.
(PDF)

**S2 Fig. (A) *Agrin* Z exon alignment. (B)** Annotated *B. lanceolatum* (amphioxus) putative *Agrin* Z exon region. Based on alignment of Augustus Gene Prediction model g13824.t1 on UCSC browser.
(PDF)

**S3 Fig. (A) *C. elegans Agrin* intron where Z exons should have been.** Based on alignment of RefSeq Gene *agr-1* on UCSC genome browser. **(B)** *S. purpuratus Agrin* intron where Z exons should have been. Based on alignment of mRNA JT097480 on UCSC genome browser.
(PDF)

**S4 Fig. RT-PCR detection of different Nova alternative splice forms in larvae (22.5 h post-fertilization) and adult brain.** M: DNA molecular weight marker in kilobase pairs.
(PDF)

**S5 Fig. Additional *Nova* expression profiling (A) UMAP plots showing re-analysis of *Nova* expression from recent single-cell RNAseq profiling of pooled cells from whole embryos collected at hourly time points spanning 5–14 hpf at 18 °C (underlying data can be found in Bernadskaya and colleagues [59]).** Left: annotated cell clusters (CNS: central nervous system, PGCs: primordial germ cells), right: *Nova* expression mapped onto cell clusters, showing enrichment in a subset of posterior mesenchyme cells that also express *Hlx* (Imai and colleagues (2004), Cao and colleagues (2019)), and in the CNS, as confirmed by in situ hybridization (see Fig 2). **(B)** *Nova* mRNA in situ hybridization coupled to immunostaining-based detection of Fgf8/17/18 > H2B::mCherry expression, revealing identity of *Nova+* MN2 cells adjacent to the *Fgf8/17/18* reporter-expressing cells.
(PDF)

**S6 Fig. (A) western blot indicating comparable expression of all three *Ciona* Nova isoforms tested, in mammalian cells.** All Nova proteins were fused to GFP ("enhanced GFP", or EGFP) at the N-terminus. Anti-GFP primary antibody was used to detect GFP::Nova isoform expression. Untagged GFP expression was monitored as a positive control. Red bands: molecular weight marker in kilodaltons (KD). See materials and methods for details. **(B)** Both "MMM" and "MLN" isoforms of *Ciona* Nova can splice *Ciona Agrin* minigenes in mammalian cells. "MEY" isoform is the same as "MLN" but using a hypothetical alternate start codon four amino acid residues after the MLN start codon. This was tested even though there is no evidence to suggest the MEY isoform is a naturally occurring one. M: DNA

molecular weight marker in kilobase pairs. $H_2O$: using water instead of cDNA template for PCR. −RT: no reverse transcriptase added.
(PDF)

**S7 Fig. GFP fluorescence images of transfected mammalian cells to monitor *Ciona* and mouse Nova expression and localization in the minigene assays.** All Nova proteins were fused to GFP ("enhanced GFP", or EGFP) at the N-terminus. Different *Ciona* Nova isoforms ("MLN", "MMM", "MEY") differed by their N-termini, thus N-terminal deletions (ΔNter) were identical regardless of isoform.
(PDF)

**S8 Fig. (A) Replicate data showing repeated *Ciona Agrin* minigene Z exon inclusion by *Ciona* Nova proteins (MEY and MLN putative isoforms), but not by mouse Nova1 or Nova2. (B)** Same minigene assay using different combinations of GDDG mutant KH domains in *Ciona* Nova "MLN" isoform. **(C)** Replicate of minigene assay using different *Ciona* Nova (MLN) deletion mutants (see Fig 3). M: DNA molecular weight marker in kilobase pairs. $H_2O$: using water instead of cDNA template for PCR. no RT: no reverse transcriptase added.
(PDF)

**S9 Fig. *Ciona Agrin* minigene assay repeated using "MMM" isoform versions of the Nova KH domain "GDDG" mutant and deletions, replicating the effects observed with the "MLN" versions (see main figures and S8 Fig).** Note: all "MMM" versions lacking the N-terminus are the same constructs as for the "MLN" versions, due to the only difference between these isoforms is the N-terminus. See Supplemental Sequences file for detailed sequence information. M: DNA molecular weight marker in kilobase pairs. $H_2O$: using water instead of cDNA template for PCR. no RT: no reverse transcriptase added.
(PDF)

**S10 Fig. Additional candidate YCAY site mutagenesis experiments.** Experiment performed, assayed, and presented as in Fig 4. Smaller products seen with exonic YCAY > YAAY mutations likely represent aberrantly-running products. M: DNA molecular weight marker in kilobase pairs. $H_2O$: using water instead of cDNA template for PCR. no RT: no reverse transcriptase added.
(PDF)

**S11 Fig. Scoring of AChRA1::GFP clustering upon initial neural-specific CRISPR-based disruption of Agrin using 6 different sgRNA expression cassettes in the form of PCR products.** Underlying data can be found in S1 Data file.
(PDF)

**S12 Fig. Illumina sequencing-based validation of selected Agrin sgRNAs.** Red arrows indicate indel plots showing estimated sgRNA efficacy. Red asterisks indicate naturally-occurring indels. Agrin sgRNAs 2 and 4 were not validated due to their inclusion in the seemingly least effective sgRNA combination, combo #2 (see Fig 5). All underlying NGS data from Azenta/Genewiz can be found at https://osf.io/xdc7t/.
(PDF)

**S13 Fig. "Peakshift" (Sanger sequencing-based) validation of Lrp4 targeting sgRNAs.** Underlying Sanger sequencing data can be found in S1 Data file.
(PDF)

**S14 Fig. "Peakshift" validation of Nova-targeting sgRNAs.** Underlying Sanger sequencing data can be found in S1 Data file.
(PDF)

**S15 Fig. Smaller fragment upstream of the "MMM" isoform start (exon 1a) does not drive neuronal expression.**
Larva electroporated with *Nova[1a] − 922/-1 > GFP,* showing fluorescence in mesenchyme and epidermis, but not in neurons.
(PDF)

**S1 Data. Supplemental data.**
(XLSX)

**S1 Raw Images. Original raw gel images.**
(PDF)

**S1 File. Supplemental sequences file.**
(DOCX)

## Acknowledgments

We are grateful to Dr. Atsuo Nishino for sharing the AChRA1::GFP construct, and to Dr. Robert B. Darnell for the mouse *Agrin* minigene construct. We thank members of our labs for critical feedback and suggestions. We thank Susanne Gibboney and Lindsey Cohen for technical assistance.

## Author contributions

**Conceptualization:** Md. Faruk Hossain, Alberto Stolfi, Lionel Christiaen, Matteo Ruggiu.

**Data curation:** Matteo Ruggiu.

**Formal analysis:** Md. Faruk Hossain, Sydney Popsuj, Burcu Vitrinel, Alberto Stolfi, Matteo Ruggiu.

**Funding acquisition:** Alberto Stolfi, Lionel Christiaen, Matteo Ruggiu.

**Investigation:** Md. Faruk Hossain, Sydney Popsuj, Burcu Vitrinel, Alberto Stolfi, Matteo Ruggiu.

**Methodology:** Matteo Ruggiu.

**Project administration:** Alberto Stolfi, Lionel Christiaen, Matteo Ruggiu.

**Resources:** Nicole A. Kaplan, Lionel Christiaen, Matteo Ruggiu.

**Supervision:** Alberto Stolfi, Lionel Christiaen, Matteo Ruggiu.

**Validation:** Md. Faruk Hossain, Sydney Popsuj, Nicole A. Kaplan, Matteo Ruggiu.

**Writing – original draft:** Md. Faruk Hossain, Alberto Stolfi, Matteo Ruggiu.

**Writing – review & editing:** Sydney Popsuj, Burcu Vitrinel, Nicole A. Kaplan, Lionel Christiaen.

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
