## [Editor Report · Decision Letter 0]

21 Jul 2025

Dear Dr Stolfi,

Thank you for submitting your revised manuscript from Review Commons entitled "A conserved RNA switch for acetylcholine receptor clustering at neuromuscular junctions in chordates" for consideration as a Research Article by PLOS Biology. Please accept my sincere apologies for the delay in getting back to you as we consulted with an academic editor about your submission.

Your manuscript has now been evaluated by the PLOS Biology editorial staff, as well as by an academic editor with relevant expertise, and I am writing to let you know that we would like to send your revised manuscript back to the original reviewers at Review Commons.

Once your full submission is complete, your paper will undergo a series of checks in preparation for peer review. After your manuscript has passed the checks it will be sent out for review. To provide the metadata for your submission, please Login to Editorial Manager (https://www.editorialmanager.com/pbiology) within two working days, i.e. by Jul 23 2025 11:59PM.

Kind regards,

Richard

Richard Hodge, PhD

rhodge@plos.org

PLOS

---

## [Decision Letter · Decision Letter 1]

14 Aug 2025

Dear Dr Stolfi,

Thank you for your patience while we considered your revised manuscript "A conserved RNA switch for acetylcholine receptor clustering at neuromuscular junctions in chordates" for publication as a Research Article at PLOS Biology. This revised version of your manuscript has been evaluated by the PLOS Biology editors, the Academic Editor and the original reviewers at Review Commons.

Based on the reviews, I am pleased to we are likely to accept this manuscript for publication, provided you satisfactorily address the remaining comment provided by Reviewer #2. In addition, please make sure to address the following data and other policy-related requests that I have provided below (A-G):

(A) We routinely suggest changes to titles to ensure maximum accessibility for a broad, non-specialist readership. In this case, we would suggest the following edit to the title, as follows. Please ensure you change both the manuscript file and the online submission system, as they need to match for final acceptance:

"Neuron-specific splicing of Agrin at neuromuscular junctions by Nova RNA-binding proteins is evolutionarily conserved in chordates"

(B) You may be aware of the PLOS Data Policy, which requires that all data be made available without restriction: http://journals.plos.org/plosbiology/s/data-availability. For more information, please also see this editorial: http://dx.doi.org/10.1371/journal.pbio.1001797

-Supplementary files (e.g., excel). Please ensure that all data files are uploaded as 'Supporting Information' and are invariably referred to (in the manuscript, figure legends, and the Description field when uploading your files) using the following format verbatim: S1 Data, S2 Data, etc. Multiple panels of a single or even several figures can be included as multiple sheets in one excel file that is saved using exactly the following convention: S1_Data.xlsx (using an underscore).

-Deposition in a publicly available repository. Please also provide the accession code or a reviewer link so that we may view your data before publication.

Figure 5D, 5F, 6E-F, 7D, 7G, S5A, S11, S12, S13, S14

(C) Please also ensure that each of the relevant figure legends in your manuscript include information on *WHERE THE UNDERLYING DATA CAN BE FOUND*, and ensure your supplemental data file/s has a legend.

(D) We require the original, uncropped and minimally adjusted images supporting all blot and gel results reported in the following Figures:

Figure 1D, 1G, 3B-F, 4B, 5B, 6B-C, S4, S6A-B, S8A-C, S9, S10

We will require these files before a manuscript can be accepted so please prepare and upload them now. Please carefully read our guidelines for how to prepare and upload this data: https://journals.plos.org/plosbiology/s/figures#loc-blot-and-gel-reporting-requirements

(E) Please ensure that your Data Statement in the submission system accurately describes where your data can be found and is in final format, as it will be published as written there.

(F) Please ensure that you are using best practice for statistical reporting and data presentation. These are our guidelines https://journals.plos.org/plosbiology/s/best-practices-in-research-reporting#loc-statistical-reporting and a useful resource on data presentation https://journals.plos.org/plosbiology/article?id=10.1371/journal.pbio.1002128

If you are reporting experiments where n ≤ 5, please plot each individual data point.

(G) Per journal policy, if you have generated any custom code during the course of this investigation, please make it available without restrictions. Please ensure that the code is sufficiently well documented and reusable, and that your Data Statement in the Editorial Manager submission system accurately describes where your code can be found.

We expect to receive your revised manuscript within two weeks.

*Published Peer Review History*

*Press*

Best regards,

Richard

Richard Hodge, PhD

rhodge@plos.org

Reviewer remarks:

Reviewer #1: The paper has been substantially improved, and the authors have responded well to my comments. I have no further comments.

Reviewer #2: I reviewed that paper for Research Common. I believe the authors have done a great job of addressing my concerns, all of which were minor. It looks like they also addressed the concerns of the other two reviewers. I have only one minor remaining comment, related to one of my suggestions and their response:

Comment: "308+ - can the authors clarify whether EBF knockdown has a selective effect on Nova vs general failure of the neurons to acquire a MN phenotype." Response: "We have been investigating this in a separate study on MN specification and differentiation in Ciona, which will be published as a preprint soon. EBF does not have a selective effect on Nova

expression, as it appears to be regulating multiple aspects of neuronal differentiation , consistent with its role as previously studied in Ciona and other organisms (e.g. Kratsios et al. 2012, Catela

et al. 2019, etc)....."

I think there needs to be a sentence addressing this issue in the present paper, because if EBF is preventing neuronal differentiation altogether, the interpretation will be quite different than if it regulating a subset of genes that includes Nova. 308+ - can the authors clarify whether EBF knockdown has a selective effect on Nova vs general failure of the neurons to acquire a MN phenotype. Perhaps this can be done in part with reference to the Kratsios and Catela papers, thereby avoiding danger to the preprint.

Reviewer #3 (Demian Burguera, identifies himself): The authors have adequately addressed all experimental issues to fully support the main conclusions of the manuscript. Therefore, I consider it suitable for publication.

---

## [Editor Report · Decision Letter 2]

31 Aug 2025

Dear Alberto,

On behalf of my colleagues and the Academic Editor, Claude Desplan, I am pleased to say that we can accept your manuscript for publication, provided you address any remaining formatting and reporting issues. These will be detailed in an email you should receive within 2-3 business days from our colleagues in the journal operations team; no action is required from you until then. Please note that we will not be able to formally accept your manuscript and schedule it for publication until you have completed any requested changes.

PRESS

Best wishes, 

Richard

Richard Hodge, PhD

rhodge@plos.org

PLOS
